# Online survey on healthcare skin reactions for wearing medical-grade protective equipment against COVID-19 in Hubei Province, China

Xiuqun Yuan[1ʘ], Huiqin Xi[2ʘ], Ye Le[3], Honglian Xu[4], Jing Wang[5], Xiaohong Meng📷[1]*, Yan Yang[2]*

1 Department of Urology, Renji Hospital, School of Medicine, Shanghai Jiaotong University, Shanghai, China, 2 Department of Nursing, Renji Hospital, School of Medicine, Shanghai Jiaotong University, Shanghai, China, 3 Department of Geriatrics, Renji Hospital, School of Medicine, Shanghai Jiaotong University, Shanghai, China, 4 Department of Colorectal Surgery, Changhai Hospital, Naval Medical University, Shanghai, China, 5 Department of Nursing, Shanghai Yangpu District Central Hospital (Yangpu Hospital, Tongji University), Shanghai, China

ʘ These authors contributed equally to this work.
* mengxiaohong@renji.com (XM); yangyan@renji.com (YY)

**Data Availability Statement:** All relevant data are within the paper and its Supporting Information files.

## Abstract

With the spread of Coronavirus Disease 2019 globally, more than 40,000 healthcare staff rushed to Wuhan, Hubei Province to fight against this threatening disease. All staff had to wear personal protective equipment (PPE) for several hours when caring for patients, which resulted in adverse skin reactions and injuries. In this study, we used an online questionnaire to collect the self-reported skin damages among the first-line medical staff in the epidemic. The questionnaire was designed by four front-line wound care nurses and then revised through Delphi consultants. Items mainly focused on the adverse skin reactions and preventive strategies. The survey was distributed through phone application from March 15th to March 20th and received 275 responses in total. The prevalence of skin reactions (212, 77.09%) was high in both head and hands. The common clinical symptoms of skin reactions were redness, device-like mark, and burning pain in face; and dryness, dermatitis, and itch/irritation in hands. Three risk factors included gender, level of protection, and daily wearing time of PPE were identified that caused skin reactions among medical staff. 150 of 275 (54.55%) participants took preventive strategies like prophylactic dressings, however, more than 75% users had little knowledge about dressings. We suggest the frontline staff strengthened the protection of skin integrity and reduced the prevalence of adverse skin reactions after professional education.

## Introduction

Series of pneumonia cases with unknown causes outbroke since December 2019 in Wuhan, Hubei Province China, later named as Coronavirus Disease 2019 (COVID-19) [1, 2]. The virus spread quickly, and all healthcare providers in China raced to Hubei Province and fought

**Funding:** The authors received no specific funding for this work.

**Competing interests:** he authors have declared that no competing interests exist.

against this threatening disease. Evidence showed the spread of COVID-19 was due to person-to-person transmission, like Severe Acute Respiratory Syndrome (SARS) in 2003 [3]. In was found that close contact without protection resulted in the infection among medical staff. Medical staff had to wear multiply personal protective equipment (PPE) including N95 masks, goggles, and protective suits to avoid hospital-acquired infection [4, 5]. Considering limited medical resource at the early stage of the pandemic, staff had to wear PPE for at least several hours. Some healthcare providers wore pull-up diapers to avoid additional waste of protective equipment. Long-time wearing PPE or diapers might cause series of skin reactions like itch, pain and acne. The integrity of the skin and related general health of medical staff was crucial to the self-prevention to fight against the COVID-19 [6, 7], the emerging high infectious disease. To provide evidence for further preventive strategies, we conducted an online survey to explore the incidence of skin reactions of healthcare providers in Hubei Province in the epidemic setting.

## Materials and methods

The study was a cross-sectional multicenter study to identify the common adverse skin reactions and related risk factors of the frontline staff fighting against COVID-19 caused by wearing PPE.

### Infection control protocols

During the outbreak, infection control protocols in China must in accordance with national guidelines for infection control protocols [8]. PPE usually included protective suit, N-95 respirator, 2-layer work caps and shoe covers, googles, disposable gloves and long-sleeve surgical gown. Physicians and nurses caring the patients wore sealed PPE after strict hand washing with trichloro hydroxyl diphenyl ether.

There were three levels of protection. Primary protection included work clothes, work caps, gowns, gloves, and surgical masks. Medical staff must wear at least the second level of protection when directly caring patients with COVID-19. Secondary protection required to wear N95 mask, protective suit, goggles or face shields besides primary protection. When facing droplets from respiratory tract like intubation, medical staff were required to wear the third level of protection that included full-scale respiratory protective equipment in addition to the secondary protection.

### Participants

Participants targeted in this research were all frontline medical staff in Hubei Province fighting against COVID-19. All the patients they cared were confirmed cases infected with COVID-19. The study was anonymous and was performed in accordance with the Declaration of Helsinki under public health emergency supervised and approved by Institutional Review Board of Renji Hospital, Shanghai Jiaotong University School of medicine. Written consent was obtained from participants electronically before the survey began.

### Tool development

**Step 1: Domain and items generation.** The research team consisted of 4 nurses specialized in wound, ostomy, and continence (WOC) from different hospitals in China. All of them worked at the frontline against COVID-19 and had at least five-year experience in wound care. After witnessing skin damages reported frequently by our colleagues, we organized an online meeting and listed possible adverse skin reactions based on our clinical observation and

literature review. Literature review focused on the risk factors of skin reactions due to medical equipment and PPE written in Chinese and English. In order to describe the prevalence and characteristics of skin injuries among medical staff and understand the prevention status, we drafted an online survey including the general information, workload, skin reactions, and preventive strategies.

In our draft, the general information included the gender, age, occupation, education level, working experience, and the grade of PPE. The workload contained working hours, self-perceived of moist and discomfort, and water intake. The adverse skin reactions consisted of all common skin reactions [9–11] like itch, acne, pain, dryness and all related clinical presentations in different locations of hands and head. Prevention part was designed to survey the participants' behavior for prevention and treatment.

**Step 2: 2-round revision.** After the research team reached an agreement on the survey, we emailed the survey to another five experts for consultants. All five experts had senior positions with at least 10-year working experience in WOC care, nursing management and nursing research. Two of five also had the working experience during the SARS pandemic. We explained our research purpose and attached the questionnaire for revision. After the first-round consultant, one expert suggested that we could use photos to illustrate the difference among four levels of moist; two experts pointed out that not all nurses had enough knowledge to differentiate among different skin injuries, and it would be better to describe the main clinical manifestation with photos instead of medical terminologies. The expert also mentioned that skin damage could occur more than face, hands, and perineal areas. Another expert suggested that how frontline nurses preferred to obtain protective information was as an indication for further staff education. We clarified the level of damage and the medical terminologies with photos and illustrations, provided more space for comments, and complemented one question regarding information source. We emailed for the second-round consultant and all experts responded within one week without further comments or revisions.

The final version of the online questionnaire consisted of 22 items. If the healthcare providers reported of related adverse skin reactions, more options like the location, clinical presentations would appear for details. Before the investigation, twenty healthcare providers with different occupations were tested as the pre-experiment. They all completed the survey within two minutes and reported no misunderstanding or confusion.

**Step 3: Survey delivery.** The survey was released through "the Questionnaire Star" website and shared with frontline healthcare providers fighting against COVID-19 through WeChat APP, the most popular chatting application in China, from March 15[th] to March 20[th]. We disseminated the electronic questionnaire to the directors of medical teams in Hubei Province. Then the link was forwarded to their medical staff within the same working group. Consent would be gained electronically before the start of the survey. Participants clicked the link and submitted the questionnaire responses electronically within one week.

## Statistical analysis

All the data were first derived from "the Questionnaire Star" website, then checked by two researchers and analyzed with SPSS 20. 0. (SPSS Inc., Chicago, IL, United States). Enumeration data was displayed with frequency and percentage. Measurement data was described with average and standard deviation. Fisher's exact or chi-square tests were applied for comparing enumerative variables, odds ratio, 95% confidence interval (95% CI). Univariate analysis was first performed for screening potential factors or skin reactions due to wearing PPE. Variables with $P$ value $<0.1$ were further analyzed by multivariate logistic regression.

## Results

A total of 275 participants in Hubei Province including 77 physicians, 197 nurses, and 1 technician were surveyed. Of the 275 participants, 65 (23.63%) were male, and 232 (84.36%) had at least Bachelor's degree. 235 healthcare providers worked in Wuhan, 36 in Xiaogan, 34 and 1 in Shiyan and Huanggang respectively. The average age was 30.7±4.34 years-old with 7±4.24 years working experience. These healthcare staff worked in Hubei province from 5 to 62 days (48±10.46). The overall prevalence of skin reactions in medical staff was 77.09% (212 of 275), and participants' characteristics were illustrated in Table 1.

### Adverse skin reactions

Adverse skin reactions happened due to long-time wearing PPE (6±1.45 hours). 215 (78.18%) participants wore PPE for over 4 hours, and longest wearing time was 10 hours (3, 1.09%). Pressure and moist was common in healthcare providers. Pressure was mostly felt under nasal bridge (216,78.54%), cheek (194, 70.55%), forehead (153, 55.63%), and auricle (144, 52.36%), which was in accordance to the locations of skin damage on face. Nasal bridge (115, 54.25%), cheek (112, 52.83%), forehead (55,25.94%) and auricle (46, 21.70%) were the most common

**Table 1. Characteristics and univariate analysis of adverse skin reactions among firstline medical staff (n = 275).**

| Characteristics | Number | Skin Reactions (n, %) | | | Prevalence of Skin Reactions (n, %) | OR[a] | 95% CI[b] | p value |
|---|---|---|---|---|---|---|---|---|
| | | Face | Hand | Both face and hand | | | | |
| **Gender** | | | | | | 3.434 | 1.967–5.966 | <0.001 |
| Male | 65 | 21(32.31) | 8(12.31) | 9(13.85) | 38(58.46) | | | |
| Female | 210 | 88(41.90) | 27(12.86) | 59(28.10) | 174(82.86) | | | |
| **Occupations** | | | | | | 2.525 | 1.465–4.354 | 0.001 |
| Physicians | 78 | 22(28.20) | 16(20.51) | 13(16.67) | 51(65.38) | | | |
| Nurses | 197 | 87(44.16) | 19(9.64) | 55(27.92) | 161(81.73) | | | |
| **Education** | | | | | | 1.577 | 0.808–3.078 | 0.179 |
| College and below | 43 | 16(37.21) | 1(2.33) | 13(30.23) | 30(69.77) | | | |
| Bachelor and above | 232 | 93(40.09) | 34(14.66) | 55(23.71) | 172(74.14) | | | |
| **Level of Protection** | | | | | | 2.037 | 1.090–3.810 | 0.024 |
| Primary | 23 | 9(39.13) | 2 | 4 | 15 | | | |
| Secondary and Third | 252 | 87 | 33 | 64 | 197 | | | |
| **Average Daily Wearing Time (PPE)** | | | | | | 1.804 | 1.018–3.198 | 0.041 |
| <4 hours | 59 | 19(32.20) | 12(20.34) | 11(18.64) | 42(71.19) | | | |
| ≥4 hours | 216 | 90(41.67) | 23(10.65) | 57(26.39) | 170(78.70) | | | |
| **Level of Moist (Protective Suit)** | | | | | | 0.779 | 0.451–1.348 | 0.372 |
| Always Moist | 113 | 38 | 15 | 37 | 90 | | | |
| Sometimes Moist | 162 | 62 | 25 | 35 | 122 | | | |
| **Level of Moist (Goggles)** | | | | | | 0.788 | 0.462–1.345 | 0.382 |
| Always Moist | 165 | 59 | 25 | 46 | 130 | | | |
| Sometimes Moist | 110 | 42 | 14 | 26 | 82 | | | |
| **Level of Moist (N-95/surgical mask)** | | | | | | 0.730 | 0.427–1.248 | 0.248 |
| Always Moist | 130 | 41 | 18 | 45 | 104 | | | |
| Sometimes Moist | 145 | 59 | 21 | 28 | 108 | | | |

Notes

a: OR: Odds Ratio.

b: CI: Confidence Interval.

**Table 2. Distribution of skin reaction manifestation in 275 medical staff (n, %).**

| Locations | numbers | Redness | Burning Pain | Dermatitis | Itch/Irritation | Device-like mark | Dryness | Blister | Injury/Breakdown | Others |
|---|---|---|---|---|---|---|---|---|---|---|
| **Face** | 177 | 129 | 107 | 17 | 52 | 113 | 0 | 29 | 46 | 5[a] |
| | | (72.88) | (60.45) | (9.60) | (29.38) | (63.84) | | (16.38) | (25.99) | (2.82) |
| **Hands** | 103 | 36 | 24 | 47 | 45 | 0 | 59 | 15 | 23 | 2[b] |
| | | (33.96) | (23.30) | (45.63) | (43.69) | | (57.28) | (14.56) | (22.33) | (1.94) |

Notes

a: acne.

b: one for limb numbness; another one for beriberi.

self-reported adverse skin reactions after wearing PPE. Moist usually existed under different PPE (see Table 1). 103 people (37.45%) found skin injuries in hand, including fingers (71,68.93%), hand back (51,49.51%), palm (28,27.18%), and waist (20,19.41%). The adverse skin reactions in face and hands were illustrated in Table 2. Redness was the most common clinical symptoms in face, and dryness in hands respectively. Of the 103 participants having hand skin reactions, 72 (69.90%) staff wore 2-layer of gloves, 25 (24.27%) wore 3-layer of gloves.

Multivariate analysis was performed between the skin reactions as the dependent variable (0 = None, 1 = Yes) and the independent variables, which were the single factors identified in Table 1 (p <0.05). The independent variables were: female = 1; nurses = 1; average daily working hours > 4 hours/day = 1. The variable "occupation", failed to present significant values and with little adherence, was removed from the multivariate logistic model with the stepwise method. Gender, level of protection and average daily wearing time of PPE remained in the model that related to the skin reactions. The $R^2$ of the final model was 0.72, indicating that 72% these included independent variables could lead to the skin reactions. Other related results were demonstrated in Table 3.

## Prevention and treatment

150 (54.55%) participants took at least one preventive strategies like prophylactic dressings, moisturizer or ointments for external use to avoid skin reactions. Prophylactic dressings were highly preferred with 109 respondents for hydrocolloid, 52 for foam dressings, and 1 for film dressing. Other preventive strategies included liquid dressing (76 participants), glycerin creams (11 participants), hormone ointments (12 participants), moisturizers (2 participants) and anti-bacteria spray (11 participants), Of the 212 participants who had skin injuries, 112 medical staff had already took preventive strategies; and of 39 participants who experienced UTI, 5 (12.82%) took medicine for treatment. However, approximately 10% participants (13, 8.67%) didn't know the effects of these preventive products, and 14% participants didn't understand the pros and cons of these products. For healthcare providers, the main sources of information for prevention were obtained from hospital training (126, 84%), recommendations from colleagues (98, 38.67%) and online (58, 38.67%).

**Table 3. Multivariate analysis of factors resulted into skin reactions in the final model.**

| Factors | B | p | Inferior | Superior |
|---|---|---|---|---|
| **Gender** | 0.139 | 0.000 | 0.066 | 0.212 |
| **Level of Protection** | 0.053 | 0.093 | -0.009 | 0.114 |
| **Daily Wearing Time of PPE** | 0.121 | 0.001 | 0.048 | 0.194 |
| **Constant** | 1.282 | 0.000 | | |

## Discussion

PPE was recommended by national guidelines [8] and World Health Organization (WHO) [2] for healthcare providers caring patients with suspected or confirmed COVID-19. Within limited resources, staff had to wear PPE and diapers for regulated time. These frontline healthcare workers are therefore susceptible to adverse skin reactions and UTS due to sustained exposure to pressure, moist and other related physical factors [6]. However, little is known about the prevalence and characteristics of PPE-related skin reactions [12]. We conducted a cross-sectional online questionnaire survey to take a deeper look at the current skin status of healthcare providers and provided evidence for further research.

The prevalence of PPE-related skin reactions was high in frontline medical staff in our research. 212 (77.09%) participants suffered different level of skin reactions, of which 44 suffered skin breakdown. Percentage of damaged skin was proved much higher than patients who had device-related pressure injury (DRPI) after wearing full-face noninvasive ventilation masks (23%) [13]. Considering some manifestations or locations of DRPI was similar to partial skin reactions to PPE, we supposed that without professional training, it was difficult for medical staff to correctly distinguish adverse skin reactions from DRPI, moist-related associated dermatitis (MASD), skin tears, or other skin injuries [14]. Responses in our research indicated the types of skin reactions were more complicated, some participants expressed multiply symptoms while some just presented with redness for minutes [6, 15]. Redness for minutes under the mask represented some kind of skin injuries, however, it was inappropriate to categorize this reaction into pressure injury. Results also confirmed that the prevalence of skin reactions in medical staff was even higher than skin injuries like DRPI and MASD in critical ill patients [13]. The most likely reason was the moist under PPE. Even the moist level of protective suit, masks, and goggles were not related to the occurrence of skin reactions, the moist increased the shear force and reduced the skin tolerance [14].

One prevalence research in Singapore during SARS outbreak confirmed that the use of PPE (N95 mask, gloves, and gown) was associated with high rates of adverse skin reactions [9]. Other reported risk factors of skin injuries after this VOCID-19 outbreak included heavy sweat, male, age over 35 years, occupation (doctors), use of prevention inputs, grade of PPE, and daily wearing time of PPE over 4 to 6 hours [15–17]. Factors such as male, level of PPE, the daily wearing time of PPE over 4 hours were also identified in our research. Male was regarded as an independent factor maybe because of their paying less attention to skin protection when treating COVID-19 infected patients [16]. 252 (91.64%) healthcare providers required the secondary and third level of protection. Level 2 and 3 PPE protection made sweat and water vapors soak and macerate the skin for long periods, which was the main difference from the first level of protection. Moist under the PPE resulted in a more susceptible microclimate to forces and shears that increased the risk of skin reactions [17].

Adherence to PPE for hours supposed to cause skin lesions associated with the constant pressure, heat and friction [9, 18]. Due to lack of medical staff and protective equipment during the COVID-19 outbreak, most staff had to work longer than four hours recommended. Considering 216 (78.55%) participants overworked, it is not surprising that the presence of skin reactions was as high as 77.09%, and in accordance with the working hours (Table 1).

This research mainly investigated the symptoms in face and hands. Redness, burning pain, and device-like mark were three common complaints in healthcare providers' face after removing the PPE (Table 3). The anatomy in face, especially nasal bridge, lacked adipose tissue, which raised the risks for skin injuries [19]. Risks factors for device-related skin injuries were similar between healthcare staff wearing PPE and patients wearing nasal-oral and full-face noninvasive ventilation masks [17]. Gas, heat, and moist under N95 masks (Table 2)

increased the friction between face and PPE, which created a hot and humid microclimate exacerbated the risks for skin reactions [15, 20]. Most skin reactions to gloves included dryness, dermatitis, and itch in our research (Table 3), which was similar to a study by Foo [9]. One possible explanation was latex sensitization, ranging from 3% to 17% among healthcare staff [21]. Medical staff had to wear gloves all the time when caring patients, whatever level of protection. Longtime wearing gloves increased the possibility of sensitivity resulted in discomfort of hand skin. Another possible reason was frequent hand-washing. Health providers had to wash at least 10 times during the process of taking off PPE, and wash hands every time after each procedure [13, 16].

Only 54.55% participants (150 of 275) took preventive strategies in advance, including moisturizers, dressings and ointments. 21 of those 150 participants had little knowledge about how to use these dressings appropriately and correctly. Information mostly came from hospital training, however, recommendations from colleagues were another main source. All dressings had unique characteristics and disadvantages, it was difficult to ensure the quality of information without professional training. Considering appropriate use of these preventive products was crucial to keep PPE sealed and skin safe [14]. Results indicated the insufficient prevention. Medical staff spared no time for the prevention during the early period; later little educational materials were available for medical staff. Education and training should be strengthened in preparation for public health emergencies despite respiratory transmitted diseases. Education should involve skin hygiene, application of sealant and skin protector to avoid skin reactions and damages. Currently there was no consensus or guidelines for self-protection, more high-quality studies were required for the safety of medical staff and patients.

## Limitations

Limitations of this research include not enough participants outside Wuhan to avoid response bias. Moreover, we just listed some common clinical symptoms of skin and urinary tract. The perceived symptoms could not be verified or diagnosed by investigators. What we reported were just the subjective assessment from participants which might cause bias as well. Lastly, we hoped to add more information from male and other healthcare providers like physicians or lab technicians to compare the difference based on the gender and occupations.

## Conclusions

Skins reactions were common in frontline healthcare providers fighting against COVID-19. To our knowledge, this has not been investigated together in other research. Skin is the first line against the physical and chemical forces under PPE. Maintaining the integrity of the skin barrier is crucial for self-protection and increase the possibility of infected with COVID-19 [17]. It is suggested that more attention should be paid to skin safety and proper preventive strategies should be taken for skin care. Some medical staff have already realized the significance of protection but without enough knowledge and skills. Any skin impairment caused by PPE should be treated immediately during the fight against the COVID-19. Currently the threat of epidemic is still alarming, our study provides the evidence of the high incidence of adverse skin reactions and hopes to promote the education of preventive strategies for healthcare fighters worldwide.

## Supporting information

**S1 Data.**
(SAV)

## Acknowledgments

We would like to thank all the frontline medical staff in Hubei Province who giving up their rest time to complete this online survey.

## Author Contributions

**Conceptualization:** Huiqin Xi, Xiaohong Meng.

**Data curation:** Huiqin Xi.

**Formal analysis:** Xiuqun Yuan.

**Investigation:** Ye Le, Xiaohong Meng.

**Methodology:** Jing Wang.

**Project administration:** Xiaohong Meng, Yan Yang.

**Resources:** Huiqin Xi, Ye Le, Honglian Xu, Jing Wang.

**Supervision:** Ye Le, Xiaohong Meng, Yan Yang.

**Writing – original draft:** Xiuqun Yuan.

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
