## [Decision Letter · Decision Letter 0]

4 Feb 2021

PONE-D-20-14972

Online survey on healthcare skin reactions and urinary symptoms for wearing medical protective equipment in Hubei Province, China

PLOS ONE

Dear Dr. Meng,

Thank you for submitting your manuscript to PLOS ONE. After careful consideration, we feel that it has merit but does not fully meet PLOS ONE’s publication criteria as it currently stands. Therefore, we invite you to submit a revised version of the manuscript that addresses the points raised during the review process.

The manuscript has been evaluated by two reviewers, and their comments are available below.

The reviewers have raised a number of concerns that need attention. They request additional information on methodological aspects of the study, revisions and additions to the statistical analyses, and they have raised concerns regarding the potential for response bias. 

Could you please revise the manuscript to carefully address the concerns raised?

We look forward to receiving your revised manuscript.

Kind regards,

Vanessa Carels

Staff Editor

PLOS ONE

5. Please amend either the title on the online submission form (via Edit Submission) or the title in the manuscript so that they are identical.

Reviewers' comments:

Reviewer's Responses to Questions

**Comments to the Author**

1. Is the manuscript technically sound, and do the data support the conclusions?

Reviewer #1: Partly

Reviewer #2: Partly

2. Has the statistical analysis been performed appropriately and rigorously? 

Reviewer #1: No

Reviewer #2: No

3. Have the authors made all data underlying the findings in their manuscript fully available?

Reviewer #1: Yes

Reviewer #2: No

4. Is the manuscript presented in an intelligible fashion and written in standard English?

Reviewer #1: Yes

Reviewer #2: No

5. Review Comments to the Author

Reviewer #1: Important text regarding the occurrence of adverse skin and urinary events due to the use of PPE.

The authors describe the problem with theoretical support.

I make considerations about the method:

1) Cross-sectional studies suggest prevalence information.

2) It´s interesting to carry out descriptive statistical tests (simple and absolute frequencies, mean, standard deviation) and to calculate the point prevalence.

3) It´s important calculating the association between explanatory variables.

4) Describe the Delph technique in detail and the number of rounds to validate the instrument.

5) The strategy for recruiting participants must be detailed in the text. How was the invitation forwarded?

6) Is it possible to describe which dressings were used for prevention? What types of moisturizing creams?

7) I suggest reading this article that deals with the same subject with similar methodology: https://www.scielo.br/scielo.php?script=sci_arttext&pid=S0034-71672020001400159&lng=pt&nrm=iso&tlng=en&ORIGINALLANG=en

Reviewer #2: The paper reports data on skin symptoms due to use of personal protective equipment in a small number of health care workers. In general, the paper reports high percentage of irritant symptoms with a very limited statistical analysis. Moreover, also the discussion is quite poor on skin symptoms due to PPE that are reported in many papers, not in relation to COVID-19 diseases.

I suggest some improvement

1. Title: Urinary symptoms. I think that put together skin symptoms with urinary symptoms can lead to misunderstanding of the reactions to personal protective equipment. I suggest to describe only skin symptoms

2. Abstract

Please delete urinary symptoms: there are not reason to put all together

3. Text The number of subjects investigated is low and positive answers for skin symptoms (more than 70%) suggests that there were a selection bias (only workers with symptoms answered the questionnaire?. Please specify the response rate of the population investigated)

4. Line 56 and urinary tract symptoms (UTS) like frequent urination due to consistent heat or prolonged voiding

Please delete

Line 134 The A. reports incidence of skin symptoms, please note that incidence are new cases on considered time, while the table reported data on prevalence of symptoms

5. Consider to do a better statistical analysis to verify factors related to skin symptoms

6. Improve discussion with data on skin symptoms in relation to masks and gloves

6. PLOS authors have the option to publish the peer review history of their article (what does this mean?). If published, this will include your full peer review and any attached files.

Reviewer #1: **Yes: **JULIANO TEIXEIRA MORAES

Reviewer #2: No

---

## [Author Response · Author response to Decision Letter 0]

3 Apr 2021

Dear Reviewers,

Thank you so much for your advice, and it really helps!

I tried my best to revise the manuscript to meet the requirements of the journal with your suggestion, and I hoped this time the manuscript would be better.

Following was my response to reviews.

Reviewer #1: Important text regarding the occurrence of adverse skin and urinary events due to the use of PPE.

The authors describe the problem with theoretical support.

I make considerations about the method:

1) Cross-sectional studies suggest prevalence information.

I amended a sentence of the overall prevalence of skin reactions in medical staff in the Results Part. 

2) It´s interesting to carry out descriptive statistical tests (simple and absolute frequencies, mean, standard deviation) and to calculate the point prevalence.

I revised my statistic methods and hoped this time it would be better. 

3) It´s important calculating the association between explanatory variables.

Thanks for your advice and I used the multivariate analysis to explore the association between explanatory variables. 

4) Describe the Delph technique in detail and the number of rounds to validate the instrument.

I described the Delphi technique more clearly in the Method part. And I validated the instrument with two-round consultants. 

5) The strategy for recruiting participants must be detailed in the text. How was the invitation forwarded?

I described more in detail with the app to recruit the patients in the Method part. 

6) Is it possible to describe which dressings were used for prevention? What types of moisturizing creams?

I described the types of dressings, and the components of the moisturizing creams.

7) I suggest reading this article that deals with the same subject with similar methodology: https://www.scielo.br/scielo.php?script=sci_arttext&pid=S0034-71672020001400159&lng=pt&nrm=iso&tlng=en&ORIGINALLANG=en

I read this article, and it really inspired me. Thank you so much.

Reviewer #2: The paper reports data on skin symptoms due to use of personal protective equipment in a small number of health care workers. In general, the paper reports high percentage of irritant symptoms with a very limited statistical analysis. Moreover, also the discussion is quite poor on skin symptoms due to PPE that are reported in many papers, not in relation to COVID-19 diseases.

I suggest some improvement

1. Title: Urinary symptoms. I think that put together skin symptoms with urinary symptoms can lead to misunderstanding of the reactions to personal protective equipment. I suggest to describe only skin symptoms

This would be a huge change so I thought it for a long time, and I agreed you were right. I deleted all the parts that related to urinary symptoms.

2. Abstract

Please delete urinary symptoms: there are not reason to put all together

I deleted all the contents about urinary symptoms

3. Text The number of subjects investigated is low and positive answers for skin symptoms (more than 70%) suggests that there were a selection bias (only workers with symptoms answered the questionnaire?. Please specify the response rate of the population investigated)

I failed to express clearly in the Method part, and I revised it to avoid misunderstanding.

4. Line 56 and urinary tract symptoms (UTS) like frequent urination due to consistent heat or prolonged voiding

Please delete

Line 134 The A. reports incidence of skin symptoms, please note that incidence are new cases on considered time, while the table reported data on prevalence of symptoms

I deleted the contents about UTS, and I changed the incidence with prevalence.

5. Consider to do a better statistical analysis to verify factors related to skin symptoms

I asked the statistical expert for help with data analysis. And tried my best to perfect the statistical analysis.

6. Improve discussion with data on skin symptoms in relation to masks and gloves

I revised my discussion part a lot and analyzed more about the reason of skin reactions due to masks and gloves.

---

## [Editor Report · Decision Letter 1]

16 Apr 2021

Online survey on healthcare skin reactions for wearing medical-grade protective equipment against COVID-19 in Hubei Province, China

PONE-D-20-14972R1

Dear Dr. Meng,

We’re pleased to inform you that your manuscript has been judged scientifically suitable for publication and will be formally accepted for publication once it meets all outstanding technical requirements.

Kind regards,

Juliano Teixeira Moraes

Guest Editor

PLOS ONE
---

## [Editor Report · Acceptance letter]

21 Apr 2021

PONE-D-20-14972R1 

Online survey on healthcare skin reactions for wearing medical-grade protective equipment against COVID-19 in Hubei Province, China 

Dear Dr. Meng:

I'm pleased to inform you that your manuscript has been deemed suitable for publication in PLOS ONE. Congratulations! Your manuscript is now with our production department. 

Kind regards, 

on behalf of

Dr. Juliano Teixeira Moraes 

Guest Editor

PLOS ONE